# Inflammation in Cerebral Cavernous Malformations: Differences Between Malformation Related Epilepsy vs. Symptomatic Hemorrhage

**DOI:** 10.3390/cells14191510

**Published:** 2025-09-27

**Authors:** Jan Rodemerk, Adrian Engel, Julius L. H. Horstmann, Laurèl Rauschenbach, Marvin Darkwah Oppong, Alejandro N. Santos, Andreas Junker, Cornelius Deuschl, Michael Forsting, Yuan Zhu, Ramazan Jabbarli, Karsten H. Wrede, Börge Schmidt, Ulrich Sure, Philipp Dammann

**Affiliations:** 1Department of Neurosurgery and Spine Surgery, University Hospital Essen, University Duisburg-Essen, 45147 Essen, Germany; adrian.engel@uk-essen.de (A.E.); julius.horstmann@uk-essen.de (J.L.H.H.); laurel.rauschenbach@uk-essen.de (L.R.); marvin.darkwahoppong@uk-essen.de (M.D.O.); nicosantos93@hotmail.com (A.N.S.); yuan.zhu@uk-essen.de (Y.Z.); ramazan.jabbarli@uk-essen.de (R.J.); karsten.wrede@uk-essen.de (K.H.W.); ulrich.sure@uk-essen.de (U.S.); philipp.dammann@uk-essen.de (P.D.); 2Department for Neuropathology, University Hospital Essen, University Duisburg-Essen, 45147 Essen, Germany; andreas.junker@uk-essen.de; 3Department of Diagnostic and Interventional Radiology and Neuroradiology, University Hospital Essen, University Duisburg-Essen, 45147 Essen, Germany; cornelius.deuschl@uk-essen.de (C.D.); michael.forsting@uk-essen.de (M.F.); 4Institute for Medical Informatics, Biometry and Epidemiology (IMIBE), University Duisburg-Essen, 45147 Essen, Germany; boerge.schmidt@uk-essen.de

**Keywords:** cerebral cavernous malformation, inflammation, immunohistochemistry, Cyclooxygenase 2, NLRP3

## Abstract

Background and Objective: Cerebral cavernous malformation (CCM) is a vascular disorder causing seizures, neurological deficits, and hemorrhagic stroke. It can be sporadic or inherited via CCM1, CCM2, or CCM3 gene mutations. Inflammation is broadly recognized as a promoter of cerebral vascular malformations. This study explores inflammatory mechanisms and differences behind CCM-related hemorrhage and epilepsy. Material and Methods: The study group comprised 28 patients, ten patients with CCM-related epilepsy, and 18 patients who clinically presented with a cerebral hemorrhage at diagnosis. All patients underwent microsurgical resection of the CCMs. Formaldehyde-fixed, paraffin-embedded tissue samples were immunohistochemically stained using a monoclonal antibody against Cyclooxygenase 2 (COX-2) (Dako, Santa Clara, CA; Clone: CX-294) and NOD-, LRR-, and pyrin domain-containing protein 3 (NLRP3) (ABCAM, Cambridge, MA, USA; ab214185). MRI and clinical data were correlated with immunohistochemical findings, and the analysis was conducted utilizing the Trainable Weka Segmentation algorithm. Results: Median CCM volume was 1.68 cm^3^ (IQR: 0.85–3.07 cm^3^). There were significantly more NLRP3-positive cells (32.56% to 91.98%; mean: 65.82%, median: 68.34%; SD: ±17.70%), compared to COX-2-positive cells (1.82% to 79.69%; mean: 45.87%, median: 49.06%; SD: ±22.56%). No correlation was shown between the volume of CCMs and a hemorrhage event (*p* = 0.13, 95% CI: 0.99–1.02). Symptomatic brain hemorrhage showed a significantly increased inflammatory enzyme upregulation from both COX-2 (*p* < 0.001) and NLRP3 (*p* = 0.009) versus patients with symptomatic CCM-related epilepsy at first diagnosis. Conclusions: Inflammatory processes in CCMs seem to be driven by broad and multiple pathways because both COX-2 and NLRP3-driven inflammatory pathways are consistently activated. As a novelty, this study showed that patients with symptomatic hemorrhage showed upregulated inflammatory enzyme activity compared to patients with CCM-related epilepsy. No direct links between NLRP3, COX-2 expression, and radiological, pathological, or preexisting patient conditions were found.

## 1. Introduction

Cerebral cavernous malformation (CCM) is a vascular disorder characterized by clusters of dilated, thin-walled capillaries without brain parenchyma intervening between the sinusoidal vessels in the brain and spinal cord [1]. These malformations can lead to seizures, neurological deficits, headaches, and even hemorrhagic stroke due to the fragile nature of the abnormal blood vessels, resulting in significant morbidity and mortality [2,3]. CCMs range from punctate to several centimeters in diameter and may occur anywhere in the central nervous system, with up to 20% located in the brainstem [4]. CCMs are either sporadic or inherited in an autosomal dominant pattern, with mutations in the CCM1 (KRIT1), CCM2 (MGC4607), or CCM3 (PDCD10) genes being the primary causes of familial cases [5,6,7,8]. Despite the generally benign nature of these lesions, complications can arise due to the risk of recurrent symptomatic hemorrhage and progressive neurological impairment [9,10]. Further, KRIT1 loss of function has already been linked to upregulation of inflammatory pathways, particularly COX-2 [11].

Recent research has highlighted the critical role of inflammation in the pathogenesis and progression of CCM. The inflammatory pathway contributes to endothelial dysfunction, exacerbating the permeability and instability of the blood vessels in CCM lesions [12,13]. Key mediators such as cytokines, including interleukins (e.g., IL-1β and IL-6), are pivotal in promoting an inflammatory environment within the cerebral vasculature [14].

Interleukin-6 has been shown to stimulate the transcription of COX-2, which is recognized as a key enzyme in the endovascular inflammation process [15,16,17]. The biochemistry of COX-2 is connected to NOD-, LRR-, and pyrin domain-containing protein 3 (NLRP3) through direct and indirect inhibition [18]. NLRP3 has been linked to autoinflammatory diseases and, more recently, to the inflammatory response to cerebral ischemia [19]. COX-2 is largely expressed due to local physical stress in the brain vessels [20]. This study aims to better understand the inflammatory mechanisms behind CCM and determine the enzymatic differences between cavernomas that present with symptomatic hemorrhage or CCM-related epilepsy.

## 2. Materials and Methods

### 2.1. Study Cohort

All patients who underwent microsurgical resection of cerebral cavernous malformation (CCMs) and from whom tissue samples were collected between 2005 and 2020 were considered eligible for this study. The study’s protocol was approved by the local ethics board. All participants or their legal representatives gave written informed consent before taking part. The study adhered to the principles outlined in the Declaration of Helsinki and complied with the Health Insurance Portability and Accountability Act (HIPAA) regulations.

### 2.2. Clinical Data Collection

Demographic, radiographic, and clinical information was prospectively collected and documented within a clinical digital system. The aggregated data included radiological images and reports, pathological findings, and clinical details pertaining to preexisting conditions, medications, comorbidities, and other pertinent risk factors.

### 2.3. Histopathological Data Collection

Pathological analysis and processing in hematoxylin-eosin and Perls’ Prussian blue stains were performed as part of the clinical routine. Histopathological basic characteristics, such as edema in the surrounding parenchyma, surrounding gliosis, and the presence and number of siderophages, were determined by a neuropathologist as part of his routine clinical diagnosis. The histopathological diagnostic reports were screened for those basic histological characteristics.

### 2.4. Immunohistochemical Staining

Sections of CCM wall tissue (1 µm) were deparaffinized and rehydrated using a series of descending alcohol concentrations, followed by cleaning with distilled water. Next, the sections were placed in boiling citrate buffer (pH 6.0) and heated in a 700 W microwave for 15 min. To inhibit endogenous peroxidase activity, the sections were then treated with 3% hydrogen peroxide in distilled water. Immunohistochemical (IHC) staining was performed using primary antibodies (COX-2: 1:50 dilution, DAKO, Clone: CX-294; NLRP3: 1:200 dilution, ABCAM, Cambridge, MA, USA, ab214185). After an overnight incubation period (over 12 h), the sections were washed with phosphate-buffered saline (PBS) and incubated with a biotinylated secondary antibody (1:200 dilution, Vector Laboratories, Burlingame, CA, USA; COX-2: BA-9200; NLRP3: BA-1000), followed by a horseradish peroxidase/streptavidin conjugate. The sections were then incubated with 3,3-diaminobenzidine as a substrate for 5 min to visualize positively stained cells. Lastly, the sections were counterstained with hematoxylin. Negative controls were included in each staining procedure to prevent false-positive results from non-specific secondary antibody binding. Positive control tissues—for COX-2 human adrenal parenchyma and for NLRP3 non-small cell lung cancer —were used to verify antibody specificity (Figure 1). A detailed staining protocol is available upon request.

### 2.5. Histological Image Analysis

Histopathological slides were digitized using an Aperio Brightfield slide scanner (Leica Microsystems, Wetzlar, Germany) at 20× and 40× magnification. In each slide, five representative areas were selected for further evaluation (the same regions were chosen in both the COX-2 and NLRP3 slides, ensuring no overlap between the two areas) using the CaseViewer software package (version 2.3 RTM, https://www.3dhistech.com/software/slide-viewer/ (accessed on 26 September 2025)) to maintain consistency and comparability. The criteria for selecting the areas were: 1. Cavernoma tissue visible 2. >90% tissue content, 3. Matching the staining intensity of the entire slide. Semi-quantitative analysis was conducted using the FIJI software package (version 2.9.0, https://imagej.net/Fiji/Downloads 1 December 2023) and the Trainable Weka Segmentation tool (doi:10.1093/bioinformatics/btx180) to classify microscopy pixels through machine learning. Images of at least 16 million pixels were divided into three categories: positive IHC signal, normal tissue, and nuclei, based on five variables per group. The percentage of IHC-positive cells was calculated relative to the total image area to quantify COX-2 and NLRP3 enzyme expression. An experienced team member reviewed each image to ensure the algorithm’s accuracy. This approach set up a two-factor analysis and a backup threshold for interpreting IHC results. The average coverage of IHC-positive cells across all five regions was calculated to estimate enzyme expression in the entire CCM.

### 2.6. Radiological Data Collection and Measurements

Magnetic resonance imaging (MRI) was performed utilizing systems with magnetic field strengths ranging from 1.0 to 3.0 Tesla (Siemens, GE, Philips, Toshiba). Pulse sequences employed included time-of-flight (TOF) and susceptibility-weighted imaging (SWI) sequences across all CCM cases. Image assessment, encompassing histological and MRI analyses, was conducted by two experienced neurovascular surgeons (PD, MDO) and a resident physician (JR) utilizing the Horos DICOM viewer (version 3.3.5, https://horosproject.org/). Morphological evaluations encompassed the measurement of the volume and presence of developmental venous anomalies (DVAs). The volume of CCM was ascertained through the measurement of the maximum diameters in multiplanar reconstructions of T2* MRI data. Dimensions of CCM were recorded in the horizontal plane (frontal and sagittal views) and the vertical plane (transverse view). The volume was calculated by multiplying the three measured dimensions and subsequently applying a correction factor of 0.5.

### 2.7. Statistical Analysis

Statistical analyses were conducted using the R and RStudio software packages (R version 4.4.1, https://cran.r-project.org/bin/windows/base/ (accessed on 16 June 2024); RStudio version 2024.12.0, https://rstudio.com/products/rstudio/download/ (accessed on 30 December 2024)) along with additional packages such as fBasic (version 4041.97), Readxl (version 1.4.5), GGPlot2 (version 4.0.0), and GGExtra (version 0.11.0). Proportions were calculated for categorical variables, and a Chi-square test was applied. For continuous variables, median values with interquartile ranges (IQR, between the 25th and 75th percentiles) or mean values with standard deviation (SD) were reported, based on whether the data were normally or non-normally distributed. The Spearman correlation was used for the entire cohort. Significant associations were then included in multivariable binary logistic regression to estimate odds ratios (OR) and 95% confidence intervals (95% CIs). The significance level (α) was set at 0.05. Missing data were handled using multiple imputations.

## 3. Results

### 3.1. Study Cohort

Our study cohort comprised 28 patients, of whom 60% were female (*n* = 17). Age ranged from 11 months to 55 years (median: 29 years, IQR: 19–36 years). One patient in the cohort presented a CCM1 mutation. The median volume of the CCM was 1.68 cm^3^ (IQR: 0.85–3.07 cm^3^) and ranged between 0.08 and 50 cm^3^. Eighteen of the 28 CCM patients presented clinically with a symptomatic hemorrhage event, while the remaining 10 patients were admitted with the primary diagnosis of cavernoma-related epilepsy without MRI signs of acute intra/extra-lesional hemorrhage. In histopathological reports of those 10 patients, individual scattered siderophages were detected in the tissue, indicating old blood breakdown. In contrast, all eighteen CCM patients with clinically symptomatic hemorrhage showed massive hemosiderin deposits and many siderophages as an indicator for an acute hemorrhage. Further, 32% (*n* = 9) of the population presented a DVA in the MRI.

In histopathological analysis, gliosis was reported in 71% of the cases, whereas 46% of the CCMs had edema surrounding the intracerebral lesion. No correlation was observed between the volume of CCMs and a hemorrhage event (*p* = 0.13, OR: 1.01, 95% CI: 0.99–1.02). Multiple cavernomas were reported in 14.2% of the cohort. Patients with numerous cavernomas showed a tendency of smaller CCMs, which presented with clinical symptoms, than patients with singular lesions (*p* = 0.13, singular: mean = 5.27 cm^3^, SD: ±11.03 cm^3^, multiple: mean: 1.39 cm^3^, SD: ±1.47 cm^3^). Correlations between the CCM volume and age, sex, body mass index, smoking, alcohol consumption, as well as the body height showed no statistical significance.

### 3.2. Histological Characteristics

The analyzed image sections show convoluted blood vessels lying one behind the other, in some cases interspersed with brain tissue, with large lumens, eccentric fibrosis, and excessive wall thickening, often with partial calcification and thrombosis (Figure 2). All individually analyzed sections show pathologically altered vessels diagnosed as CCM in the majority of each image section. However, the surrounding brain tissue is also visible in almost all images. Differences to healthy brain tissue in the surrounding tissue are, for example, gliosis (shown in 71% of the cases) and edema in the surrounding tissue (in 46% of the cases).

### 3.3. CCM and Its Association with NLRP3

Expression of NLRP3 in the CCM wall samples ranged from 32.56% to 91.98% (mean: 65.82%, median: 68.34%; SD: ±17.70%, Figure 3). There was no correlation between NLRP3 expression and CCM volume (*p* = 0.89). Additionally, CCMs with DVAs showed no link to increased NLRP3 enzyme expression (*p* = 0.90, OR: 0.99, 95% CI: 0.988–1.010). Even subgroup analysis of clinical symptomatic hemorrhage events (*p* = 0.34) or CCM-related epilepsy (*p* = 0.49) was not linked to higher NLRP3 expression rates in CCMs with DAVs. Histopathological characteristics of the examined tissue, like gliosis (*p* = 0.81, OR: 1.00, 95% CI: 0.990–1.011) and edema in the adjacent parenchyma (*p* = 0.26, OR: 1.01, CI: 0.995–1.017), showed no impact on the NLRP3 expression. The patient’s age showed no correlation to the NLRP3 expression (*p* = 0.87). Further, patients with multiple CCMs showed no difference in the expression rates of NLRP3 compared to those with only one CCM (*p* = 0.60). Female sex, or gender in general, also showed no impact on the enzymatic expression of NLRP3 (*p* = 0.93). Also, the body mass index showed no correlation to COX-2 (*p* = 0.64) and body height (*p* = 0.14). In addition, no correlation between clinical characteristics like hypertension, regular medication, smoking, regular alcohol intake, or drug intake could be reported to a change in NLRP3 expression patterns. However, there was a correlation between higher NLRP3 expression rates and a clinical symptomatic hemorrhage event (*p* = 0.002, OR: 1.015, 95% CI: 1.005–1.024). Even radiologically reported hemorrhage but clinically silent events showed a high tendency to higher NLRP3 expression (*p* = 0.051, OR: 1.01, 95% CI: 0.999–1.020). In turn, CCM-related epilepsy was negatively linked to higher enzymatic expression of NLRP3 (*p* = 0.01, OR: 0.98, 95% CI: 0.976–0.996).

### 3.4. CCM and Its Association with COX-2

Expression of COX-2 in the CCM wall samples varied between 1.82% and 79.69% (mean: 45.87%, median: 49.06%; SD: ±22.56%). There was no correlation between COX-2 expression and the CCM volume (*p* = 0.73, Figure 4). Furthermore, CCMs with DVAs demonstrated no correlation with an increase in COX-2 enzyme expression (*p* = 0.87, OR: 1.00, 95% CI: 0.992–1.009). Even subgroup analysis of clinical symptomatic hemorrhage events (*p* = 0.97) or CCM-related epilepsy (*p* = 0.88) was not linked to higher COX-2 expression rates in CCMs with DAVs. Histopathological characteristics of the examined tissue, like gliosis (*p* = 0.59, OR: 0.997, 95% CI: 0.989–1.006) and edema in the adjacent parenchyma (*p* = 0.78, OR: 1.001, 95% CI: 0.992–1.010) showed no impact on the COX-2 expression. The patients’ individual age showed no correlation to the COX-2 expression (*p* = 0.81). Further, patients with multiple CCMs showed no difference in the expression rates of COX-2 versus those with only one CCM (*p* = 0.35). Female sex, or gender in general, also showed no impact on the enzymatic expression of COX-2 (*p* = 0.55). Also, the body mass index showed no correlation to COX-2 (*p* = 0.88) and body height (*p* = 0.36). In addition, no correlation between clinical characteristics like hypertension, regular medication, smoking, regular alcohol intake, or drug intake could be reported to a change in COX-2 expression patterns. However, there was a correlation between higher COX-2 expression rates and a clinical symptomatic hemorrhage event (*p* < 0.001, OR: 1.013, 95% CI: 1.006–1.020). Even radiologically reported hemorrhage but clinically silent events significantly correlated to higher COX-2 expression (*p* = 0.001, OR: 1.012, 95% CI: 1.005–1.019). In turn, CCM-related epilepsy was linked to lower enzymatic expression of COX-2 (*p* = 0.03, OR: 0.991, 95% CI: 0.982–0.999, Figure 5).

### 3.5. Relationship Between the Expressed Inflammatory Enzymes in CCMs

Overall, COX-2 is expressed at approximately 66.67% of the level of NLRP3 (*p* = 0.01). For instance, patients presenting with symptomatic hemorrhage exhibited an average of 56.06% COX-2 expression, whereas NLRP3 expression averaged 72.88%. Furthermore, patients with epilepsy related to CCMs demonstrated mean expression levels of 38.17% for COX-2 and 58.66% for NLRP3. A statistically significant correlation between elevated NLRP3 and concomitant increased COX-2 expression was observed (*p* = 0.01). This expression pattern and characteristic are consistent across all variables examined in the study (see Table 1).

## 4. Discussion

This study aimed to evaluate the impact of inflammatory pathways and their histopathological markers (COX-2 and NLRP3) on the development and clinical presentation of CCMs. Histopathologically, the tissue removed during cavernoma surgery was further processed and examined morphologically and immunohistochemically. The analyzed image sections show convoluted back-to-back, but in some cases also interspersed with brain tissue, large-lumen blood vessels with eccentric fibrosis and excessive wall thickening, often with partial calcification and thrombosis. Thus, not only the CCM tissue but also the surrounding brain tissue and microenvironment are analyzed via the semiquantitative method. The complete image sections were included in the analysis in order to visualize possible changes related to the primary endpoints (CCM-related epilepsy vs. hemorrhage) not only in the CCM but also in the surrounding microenvironment. The histomorphology was presented in Figure 2 using HE staining as a reference for the analyzed tissue.

In accordance with a previous publication, we identified a consistent and elevated expression of COX-2 in the tissue of CCMs [13]. Moreover, the mostly ischemia-induced expression pattern of NLRP3 could be reproduced in our patients’ samples, as previously shown in an earlier publication [12]. As a novum, we were able to report the enzymatic differences between CCMs that presented with symptomatic hemorrhage and those that presented with cavernoma-related epilepsy. Cavernomas with symptomatic hemorrhage had a more substantial expression rate of NLRP3 and COX-2. In contrast, the CCMs with an epilepsy onset showed lower rates of both inflammatory enzymes.

Genetic mutations (such as CCM1, CCM2, and CCM3) play a decisive role, especially in patients with multiple cavernomas. Particularly in the context of driven inflammatory pathways, including increased COX-2 levels, it has been reported based on the loss-of-function mutation in the CCM genes [11,21]. Due to the lack of statistical evaluation options with the number of genetic mutation cases (*n* = 1), no statement can be made in this study regarding genetic mutations and inflammatory processes.

Patients with CCM-associated epilepsy exhibited lower expression levels of COX2 and NLRP3 compared to those with hemorrhagic CCMs; however, expression was still present. This expression also extends into the surrounding parenchyma, which, as in all CCMs, appears gliotically altered. A chronically increased inflammatory potential could lead to possible stimulus conduction problems and, thus, epileptogenic potentials in the long term. Reactive astrocyte-driven gliosis has already been linked to epileptogenesis [22]. In turn, patients with symptomatic hemorrhage showed even higher inflammatory responses, probably attributed to acute immune reactions in the tissue. The immunohistochemical data collected here do not allow the duration of the existing inflammatory cascade in the analyzed tissue to be determined. Nevertheless, due to the homogeneous changes between the classes of patients with symptomatic hemorrhage and CCM-related epilepsy, an acute reaction seems likely to be the reaction or potentially even the cause of a hemorrhage.

DVAs are frequently observed in sporadic CCM cases and have been proven to be implicated in lesion formation via inflammatory pathways [23,24]. However, our study did not detect any statistically different enzyme activity of COX-2 or NLRP3 in CCMs with DVAs compared to CCMs without DVAs. Even subgroup analysis of CCMs with symptomatic hemorrhage or CCMs with related epilepsy showed no higher enzyme activity in CCMs with DVAs. One possible reason for the lack of evidence here may be the bias of the intraoperatively obtained samples, as complete processing of the CCM is limited due to coagulation necrosis caused by the surgical technique.

COX-2 and NLRP3 inflammatory enzymes share a complex relationship in their biochemistry. NLRP3 is activated and regulated by various factors and pathways [18]. In cerebral pathophysiology, ischemia and post-ischemic reorganization are the main triggers for activating this enzyme [25]. In contrast, COX-2 activation in the cerebral vasculature is most likely due to shear stress and inflammatory cells [20]. However, both enzymes respond to a range of stimuli, including oxidative stress, metabolic changes, and molecular patterns associated with pathogens or damage [17,18,20,21,26]. Furthermore, research has shown that COX-2 has a regulatory effect on the NLRP3 inflammasome’s output [27]. Additionally, COX-2’s catalytic product, prostaglandin E2, directly inhibits the NLRP3 inflammasome [28]. This connection suggests an anticipatory relationship between these two enzymes. The collected data here cannot support a direct anticipatory relationship in the micro-milieu of CCMs. We see a substantial, nearly linear correlation between increased COX-2 and NLRP3 expression (Figure 4). This correlation emphasizes the possibility of further and more broadly activated inflammatory pathways in the pathophysiology of CCMs [29]. This positive activation connection from the NLRP3-inflammasome to the COX-2/mPGES-1/PGE2 axis we observed in our cohort has also been reported in other publications [30].

Considering these aspects, these data might back up that COX-2 and NLRP3 are essential for growth and hemorrhage events because both enzymes are strongly expressed in CCMs regardless of size. Most likely due to the minor sample size and largely healthy and young cohort, we did not detect a substantial correlation to clinical parameters like preexisting conditions, nor could we address increased expression to a radiological characteristic. Nevertheless, our results suggest a relevant role of the inflammatory pathways in the pathophysiology of CCMs and the emergence of CCM-related epilepsy and symptomatic cerebral hemorrhages. However, direct correlations of the intensity of this inflammation with clinical or radiographic factors were absent in our cohort. Ultimately, these absent correlations between inflammation and clinical characteristics in CCMs are consistent with the existing literature [31].

## 5. Limitations

All our samples came from surgically treated patients, which may introduce unavoidable selection bias to our results. Additionally, the small sample size limits the study’s statistical power. Further, our data cannot answer whether an increased inflammatory reaction occurred before or after the bleeding event. Further, despite the fact that the antibodies used are already well established, there is a possibility that variation in sensitivity and specificity of the first antibodies may alter the reproducibility and reliability of this study. In addition, IHC alone may not be able to provide information about the real functional activity of COX-2 and NLRP3. Still, the method used in this study can show a tendency toward possible functional activity. Furthermore, this study is limited by its methodology, as no cell-type-specific markers were used to determine possible differences in the analyzed tissue sections between individuals. This work is also limited by a non-homogeneous collective in terms of CCM localization, type and duration of epileptic seizures and time between surgery and cerebral hemorrhage. Nevertheless, our results demonstrate consistent enzyme expression of NLRP3 and COX-2, making the findings valuable despite the mentioned limitations.

## 6. Conclusions

The expression rate of the inflammatory enzymes COX-2 and NLRP3 in CCMs showed a distinct difference. The novelty of this study is the direct comparison of inflammatory markers in patients with symptomatic hemorrhage and CCM-related epilepsy. This study identified upregulated enzyme expression of both COX-2 and NLRP3 in patients who had a cerebral hemorrhage as the first symptom compared to patients with CCM-related epilepsy. The upregulated inflammatory pathways of COX-2 and NLRP3 in CCM tissue shown here underline the central role of inflammation in the pathophysiology of CCMs. The expression of inflammatory markers exhibited no correlation with the radiological or patient characteristics of this cohort. The specific influence of inflammatory pathways on the development and clinical progression of CCMs remains uncertain. Consequently, we strongly recommend the creation of a multicenter registry encompassing histopathological samples along with corresponding radiological and clinical data to facilitate further research into the pathophysiology of CCMs within larger cohorts.

## Figures and Tables

**Figure 1 cells-14-01510-f001:**
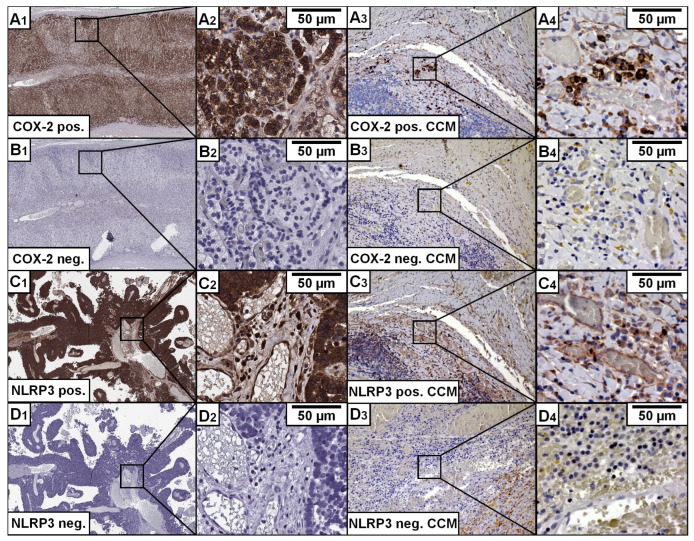
Positive and negative control for primary Antibodies. Immunohistochemically (IHC)-staining of the inflammatory Enzyme COX-2 and NLRP3 is displayed in (**A**–**D**) (X1/3 images: ×4 magnification; X2/4 images: ×40 magnification). (**A**) COX-2 Antibody with Adrenal tissue to validate the staining process for the antibody (**A1** + **A2**) and example of COX-2 staining in CCMs (**A3** + **A4**). A strong signal is depicted in the positive control. (**B**) Negative control without the first antibody shows no signal in Adrenal Tissue (**B1** + **B2**) and CCM tissue (**B3** + **B4**). (**C**) Non-small cell lung carcinoma tissue was used to validate the NLRP3 staining process. A powerful signal is shown in the positive control (**C1** + **C2**) and an example of NLRP3 staining in CCMs (**C3** + **C4**). (**D**) Negative control shows no signal without the first antibody in Non-small cell lung carcinoma tissue (**D1** + **D2**) and CCM tissue (**D3** + **D4**).

**Figure 2 cells-14-01510-f002:**
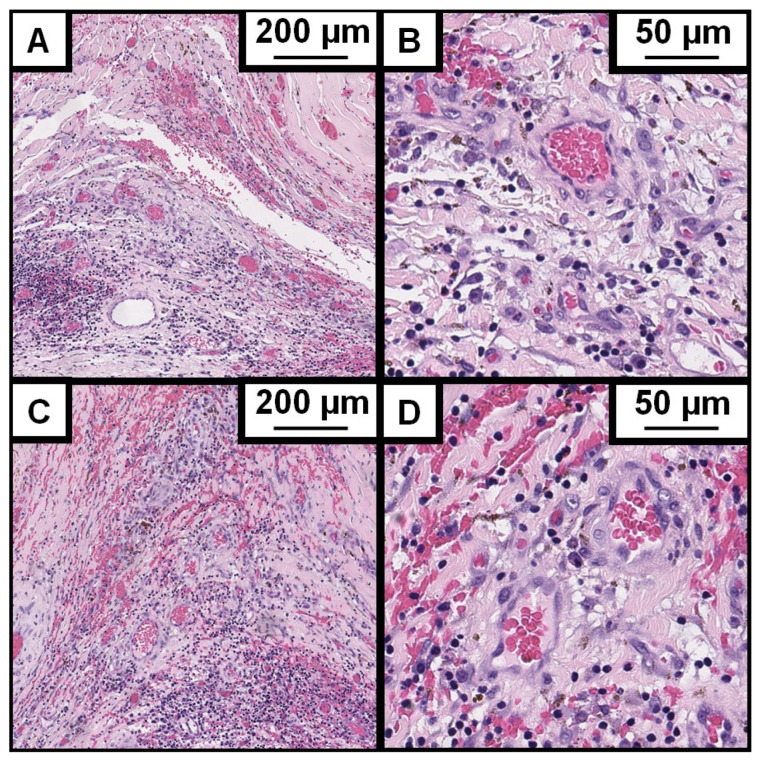
Histomorphology of analyzed CCMs. Hematoxylin and Eosin stains of cerebral cavernous malformation (CCM). Exemplary sections are displayed. (**A**) ×4 magnification; (**B**) ×40 magnification of section A. (**C**) ×4 magnification; (**D**) ×40 magnification of section C.

**Figure 3 cells-14-01510-f003:**
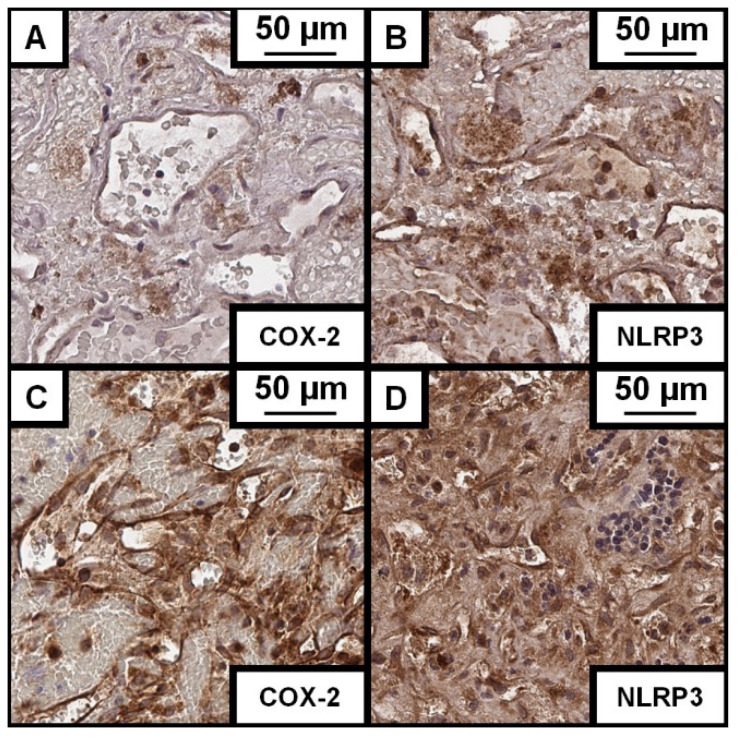
COX-2 and NLRP3 expression in CCMs. Immunohistochemical staining in 40× magnification with Cyclooxygenase 2 (COX-2) and pyrin domain-containing protein 3 (NLRP3) antibodies of cerebral cavernous malformation (CCMs). In all CCMs, an average of 65.82% of cells test positive for NLRP3. Meanwhile, an average of 45.87% of cells test positive for COX-2. (**A**) This CCM from a 20-year-old woman with a volume of 1.18 cm^3^ has a mean COX-2 expression of 46.20%, whereas the NLRP3 expression reaches % on average of 56.60% (**B**). (**C**) This CCM of a 24-year-old male had a volume of 3.21 cm^3^ and an average COX-2 expression of 79.68%. The NLRP3 enzyme expression reached 91.32%, the second highest recorded in this cohort (**D**).

**Figure 4 cells-14-01510-f004:**
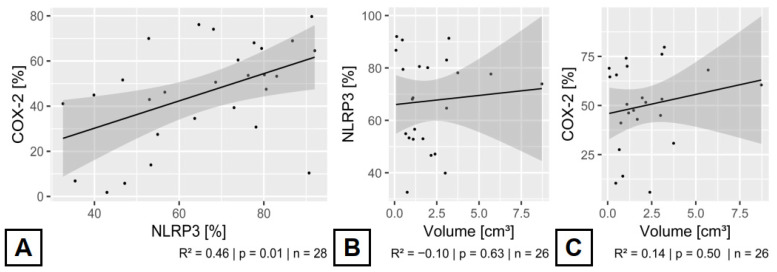
Semiquantitative Enzymatic Expression rates and their correlation to CCM volume. (**A**) Direct comparison of NOD-, LRR- and pyrin domain-containing protein 3 (NLRP3) and Cyclooxygenase 2 (COX-2) expression profiles in the wall of cerebral cavernous malformation (CCM). A positive linkage between higher enzymatic expression of COX-2 and NLRP3 could be reported (*p* = 0.01). In turn, CCM volume did not influence the enzymatic expression of COX-2 ((**B**) *p* = 0.73) and NLRP3 ((**C**) *p* = 0.78). Outliers over 20 cm^3^ (*n* = 2) were excluded in graphs (**B**,**C**) to improve the visualization.

**Figure 5 cells-14-01510-f005:**
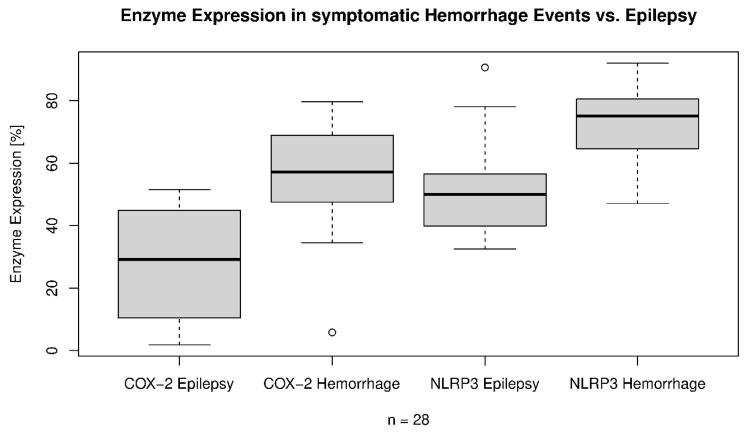
Semiquantitative Enzyme Expression in symptomatic hemorrhage events vs. CCM-related epilepsy. Patients with a first symptomatic cerebral cavernous malformation (CCM) related epilepsy had, on average, a Cyclooxygenase 2 (COX-2) expression of 27.52%, whereas the patients with a symptomatic brain hemorrhage presented on average with 56.06% (*p* < 0.001). The NOD-, LRR- and pyrin domain-containing protein 3 (NLRP3) expression in patients with a symptomatic brain hemorrhage reached an average of 72.88%, whereas patients with CCM-related epilepsy had on average only 53.11% of NLRP3 expression (*p* = 0.009).

**Table 1 cells-14-01510-t001:** Enzyme expression and patients’ characteristics. Values are presented as *p*-values, Odds-Ratios (OR) and 95% confidence interval (95% CI). The first two columns display the correlation between enzyme expression and the clinical variable. The last two columns compare NLRP3 and COX-2 directly in subgroups with and without the clinical manifestation. Values are shown as mean ± standard deviation, statistical significant variables are marked in bold. Abbreviations: COX-2, Cyclooxygenase-2; NLRP3, NOD-, LRR- and pyrin domain-containing protein 3; BMI, Body-mass-index; CCM: Cerebral cavernous malformation, DVA: developmental venous anomalies.

Manifestation	NLRP3	COX-2	NLRP3 vs. COX-2 With Manifestation	NLRP3 vs. COX-2 Without Manifestation
Pathology				
Gliosis	*p* = 0.81, OR: 1.001, 95% CI: 0.991–1.012	*p* = 0.59, OR: 0.997, 95% CI: 0.989–1.006	***p* = 0.001** 66.3 ± 18.2% vs. 44.4 ± 23.0%	*p* = 0.16 64.5 ± 17.5% vs. 49.5 ± 22.4%
Edema in adjacent parenchyma	*p* = 0.26, OR: 1.001, 95% CI: 0.995–1.018	*p* = 0.78, OR: 1.001, 95% CI: 0.992–1.010	***p* = 0.003** 69.9 ± 17.8% vs. 47.1 ± 17.4%	***p* = 0.04** 62.2 ± 17.3% vs. 44.7 ± 26.8%
Clinical data				
Age	*p* = 0.87, OR: 0.981, 95% CI: 0.716–1.345	*p* = 0.81, OR: 1.039, 95% CI: 0.812–1.331	Patients over 29 years: *p* = 0.10 61.7 ± 17.1% vs. 48.8 ± 21.8%	Patients under 30 Years: ***p* = 0.002** 69.3 ± 18.0% vs. 43.2 ± 23.5%
BMI	*p* = 0.64, OR: 0.69, 95% CI: 0.36–1.34	*p* = 0.054, OR: 0.64, 95% CI: 0.406–1.010	---///---	---///---
Body Height	*p* = 0.14, OR: 1.46, 95% CI: 0.89–2.42	*p* = 0.06, OR: 1.39, 95% CI: 0.97–2.00	---///---	---///---
Sex	*p* = 0.93, OR: 0.999, 95% CI: 0.988–1.011	*p* = 0.50, OR: 1.002, 95% CI: 0.994–1.011	Female: ***p* = 0.006** 65.5 ± 17.6% vs. 48.2 ± 17.3%	Male: ***p* = 0.03** 66.1 ± 18.6% vs. 42.2 ± 29.5%
Symptomatic hemorrhage	***p* = 0.002, OR: 1.015, 95% CI: 1.005–1.024**	***p* = < 0.001, OR: 1.013, 95% CI: 1.006–1.020**	***p* = 0.003** 72.8 ± 12.9% vs. 56.0 ± 18.0%	***p* = 0.006** 53.1 ± 18.5% vs. 27.5 ± 18.2%
CCM related epilepsy	***p* = 0.01, OR: 0.986, 95% CI: 0.976–0.996**	***p* = 0.03, OR: 0.991, 95% CI: 0.982–0.999**	***p* = 0.01** 58.6 ± 18.9% vs. 38.1 ± 25.4%	***p* < 0.001** 75.3 ± 10.3% vs. 57.1 ± 12.8%
Multiple CCMs	*p* = 0.60, OR: 0.997, 95% CI: 0.989–1.005	*p* = 0.41, OR: 1.002, 95% CI: 0.996–1.008	*p* = 0.66 60.7 ± 19.8% vs. 54.6 ± 17.6%	***p* < 0.001** 66.6 ± 17.6% vs. 44.4 ± 23.2%
Radiology				
Volume	*p* = 0.89, OR: 1.069, 95% CI: 0.82–1.39	*p* = 0.82, OR: 0.97, 95% CI: 0.78–1.21	---///---	---///---
Radiographic hemorrhage	*p* = 0.051, OR: 1.010, 95% CI: 0.999–1.021	***p* = 0.001, OR: 1.012, 95% CI: 1.005–1.019**	***p* = 0.01** 70.6 ± 14.9% vs. 55.5 ± 18.2%	***p* = 0.004** 57.1 ± 19.7% vs. 28.4 ± 19.3%
DVA	*p* = 0.90, OR: 0.99, 95% CI: 0.988–1.010	*p* = 0.87, OR: 1.00, 95% CI: 0.992–1.009	*p* = 0.044 65.2 ± 14.9% vs. 46.8 ± 20.1%	*p* = 0.006 66.0 ± 19.2% vs. 45.3 ± 24.1%

## Data Availability

The data presented in this study are available upon request from the corresponding author, as the complete data cannot be made available due to privacy restrictions.

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
