# Peer review of "Inflammation in Cerebral Cavernous Malformations: Differences Between Malformation Related Epilepsy vs. Symptomatic Hemorrhage"

_cells, 2025, doi:10.3390/cells14191510_

Round 1

Reviewer 1 Report

Comments and Suggestions for Authors

In the manuscript titled “Inflammation in Cerebral Cavernous Malformations: Differences between Malformation Related Epilepsy and Symptomatic Hemorrhage,” Rodemerk J. et al. conducted a descriptive study examining the relationship between the characteristics of human cerebral cavernous malformations (CCMs) and immunohistochemical staining for two inflammatory markers, COX-2 and NLRP3. The authors identified significant heterogeneity in the immunodetection of COX-2 and NLRP3 across the different human CCM lesions analyzed. Notably, they found that the most significant correlation between the inflammatory markers was present in samples with hemorrhagic CCM.

Few comments:

  1. The study would benefit from including clear labels for each staining directly in the figure.
  2. It is unclear how gliosis and edema were determined.
  3. It would be helpful if the authors could comment on the types of cells expressing the inflammatory markers.
  4. In Table 1, the authors provided p-values for the comparison of NLRP3 vs. COX2 in relation to manifestations. However, the meaning of these p-values is not clearly explained, especially since the markers do not show statistical significance, for instance, in relation to gliosis.
  5. It would also be very informative if the authors could elaborate more on which samples they did not detect inflammatory markers in their cohort. As it read, it seems as if the authors always found the presence of inflammatory markers, but on some occasions, the expression was low, while in others it was elevated.

Author Response

In the manuscript titled “Inflammation in Cerebral Cavernous Malformations: Differences between Malformation Related Epilepsy and Symptomatic Hemorrhage,” Rodemerk J. et al. conducted a descriptive study examining the relationship between the characteristics of human cerebral cavernous malformations (CCMs) and immunohistochemical staining for two inflammatory markers, COX-2 and NLRP3. The authors identified significant heterogeneity in the immunodetection of COX-2 and NLRP3 across the different human CCM lesions analyzed. Notably, they found that the most significant correlation between the inflammatory markers was present in samples with hemorrhagic CCM.

Few comments:

The study would benefit from including clear labels for each staining directly in the figure.

Response: We have adjusted Figures 1 and 2 accordingly and improved the cavernoma tissue pictures in Figure 1.

It is unclear how gliosis and edema were determined.

Response: Thank you for pointing out the lack of methodology for this specific data collection. Neuropathologists collected this data as part of their routine clinical work after the tissue was removed during surgery and prepared for diagnostic purposes. These data arrive from pathological reports of This data is therefore taken from the histopathological diagnostic report. Now we specified this further in the section “Methods/Histopathological data collection”.

It would be helpful if the authors could comment on the types of cells expressing the inflammatory markers.

Response: Thank you for your suggestion to improve the readability and interpretability of the results shown here. Based on this, we have discussed the morphology and analyzed cells in detail in the first section of the discussion to ensure better understanding and reproducibility.

In Table 1, the authors provided p-values for the comparison of NLRP3 vs. COX2 in relation to manifestations. However, the meaning of these p-values is not clearly explained, especially since the markers do not show statistical significance, for instance, in relation to gliosis.

Response: Thank you for pointing out the possible misunderstanding of those two table columns. As stated in the last result paragraph, those values represent the overall characteristic, that NLRP3 is more highly expressed than COX-2 (NLRP3 vs. COX-2 ~ each variable). For better readability, we provide semiquantitative expression rates in the third and fourth columns of Table 1.

It would also be very informative if the authors could elaborate more on which samples they did not detect inflammatory markers in their cohort. As it read, it seems as if the authors always found the presence of inflammatory markers, but on some occasions, the expression was low, while in others it was elevated.

Response: This is also a very interesting approach. However, as you already mentioned, all cavernoma samples show a clear expression of both inflammatory markers. Only two samples show expression below 10%, and this is only for COX-2. The characteristics of the two associated patients are very heterogeneous. There are no further correlations beyond those already mentioned regarding low enzyme expression and clinical characteristics.

Thus, due to the lack of clinical relevance and, above all, the very small sample size of tissue samples with low enzyme expression, no conclusive results can be presented in this regard.

Thank you very much for your time and effort in reviewing this work.

Reviewer 2 Report

Comments and Suggestions for Authors

Peer Review Report

Manuscript Title: Inflammation in cerebral cavernous malformations: Differences between malformation related epilepsy vs. symptomatic hemorrhage
Journal: Cells
Manuscript Number: 3810089

General Comments

This manuscript explores the role of inflammation in the clinical manifestations of cerebral cavernous malformation (CCM) disease, with a focus on its potential contributions to CCM-related cerebral hemorrhage and epilepsy. Using immunohistochemical (IHC) analysis of surgical specimens, the study examines the expression of two inflammation-associated proteins: COX-2 and NLRP3.

While the topic is timely and of potential clinical relevance, the study's overall contribution is limited by significant methodological and interpretative issues. Although the authors acknowledge some limitations, several others (both technical and conceptual) remain unaddressed. These issues collectively compromise the robustness of the findings and the strength of the conclusions.

Overall Assessment

The manuscript reiterates the already well-established role of inflammation in CCM pathogenesis, and while it may provide additional support to this model, the experimental approach and data interpretation are superficial. Furthermore, several key aspects—including appropriate controls, proper contextualization within the existing literature, and mechanistic insights—are either lacking or insufficiently developed.

At this stage, I do not recommend the manuscript for publication. A substantially revised version would be necessary, one that addresses the following major concerns in both experimental design and data interpretation.

Major Concerns

  1. Genetic and Radiological Context Lacking
    The manuscript does not specify whether any of the cases—especially those with multiple cavernomas (14.2% of the cohort)—represent familial forms associated with known mutations in CCM1, CCM2, or CCM3. This is a significant omission, as loss-of-function mutations in these genes are known to drive inflammatory signaling, including COX-2 upregulation (PMID: 24291398; 27639680).
    Furthermore, no reference is made to developmental venous anomalies (DVAs), which are frequently observed in sporadic CCM cases and have been implicated in lesion formation via inflammatory pathways (PMID: 19833796; 36498972). Including these clinical and radiological aspects would provide important context for interpreting the inflammatory findings.
  2. Antibody Validation and Reproducibility
    The differences observed in COX-2 and NLRP3 protein expression may be influenced by varying sensitivity and specificity of the primary antibodies used. The authors do not address this possibility. Validation using additional antibodies or complementary techniques (e.g., Western blotting or qPCR) is recommended to confirm the reproducibility and reliability of the IHC results.
  3. Lack of Cell-Type-Specific Analysis
    The IHC quantification appears to be performed over entire tissue sections, without differentiating between distinct cell populations (e.g., endothelial cells, glial cells, inflammatory cells). This limits the biological relevance of the data. A cell-specific analysis, possibly using co-staining or segmented image analysis, would enhance the interpretability of protein expression patterns and their functional implications.
  4. Inappropriate Interpretation of Semi-Quantitative IHC
    The IHC analysis conducted only allows for semi-quantitative assessment of protein expression. Therefore, references to enzymatic expression rates (g., Figure 3) or inflammatory pathway activation are not only inappropriate but also potentially misleading.
    In particular, NLRP3 is not an enzyme but a component of the inflammasome complex, whose activation involves intricate regulatory mechanisms, including NLRP3 post-translational modifications and interactions with other proteins, such as ASC and caspase-1, which ultimately leads to cytokine activation (e.g., IL-1β, IL-18; PMID: 36741414).
    Similarly, COX-2 has complex, context-dependent roles in both physiological and pathological processes (PMID: 9737710), and IHC alone does not provide insights into its functional activity or downstream effects.
    Thus, without measuring the levels of downstream effectors, such as COX-2-derived prostanoids (e.g., prostaglandin E2) or NLRP3-inflammasome-related cytokines (e.g., IL-1β, IL-18), the conclusions drawn about inflammatory pathway activation are speculative and unsupported.
  5. Oversimplified and Unsupported Conclusions
    The conclusion that “the upregulated inflammatory pathways suggest a dominant role of both shear stress (COX-2) and ischemia-induced inflammation (NLRP3)” (Section 6, lines 313–315) is speculative and reductive. Both COX-2 and NLRP3 are responsive to a wide array of stimuli, including oxidative stress, metabolic shifts, and pathogen- or damage-associated molecular patterns. Assigning their activation solely to shear stress or ischemia, in the absence of mechanistic evidence, is an overinterpretation.
  6. Overlooked Crosstalk Between COX-2 and NLRP3
    The manuscript treats COX-2 and NLRP3 as independent inflammatory markers, yet increasing evidence indicates a reciprocal relationship between them. COX-2 can influence NLRP3 inflammasome activation, and NLRP3 signaling can, in turn, regulate COX-2 expression (e.g., PMID: 25294243; 39832004; 28628921). This crosstalk is mechanistically relevant and should be acknowledged in the discussion, as it may affect the interpretation of their co-expression in CCM tissues.
  7. Failure to Cite Prior Foundational Studies
    The upregulation of COX-2 in CCM lesions has been originally demonstrated and discussed in previous reports (PMID: 24291398; 27639680), yet these studies are not cited. Proper citation of foundational literature is essential to acknowledge prior work and to contextualize the current findings within the evolving understanding of inflammation in CCM pathology.

Additional Comments

  • Consider including additional controls or validation experiments to strengthen the reliability of the immunohistochemical findings.
  • Ensure that all references to molecular pathways and protein functions are consistent with current literature and mechanistic understanding.
  • Including a schematic diagram summarizing the hypothesized inflammatory mechanisms (based on both current and prior findings) might help readers visualize the proposed model more effectively.

Author Response

General Comments

This manuscript explores the role of inflammation in the clinical manifestations of cerebral cavernous malformation (CCM) disease, with a focus on its potential contributions to CCM-related cerebral hemorrhage and epilepsy. Using immunohistochemical (IHC) analysis of surgical specimens, the study examines the expression of two inflammation-associated proteins: COX-2 and NLRP3.

While the topic is timely and of potential clinical relevance, the study's overall contribution is limited by significant methodological and interpretative issues. Although the authors acknowledge some limitations, several others (both technical and conceptual) remain unaddressed. These issues collectively compromise the robustness of the findings and the strength of the conclusions.

Overall Assessment

The manuscript reiterates the already well-established role of inflammation in CCM pathogenesis, and while it may provide additional support to this model, the experimental approach and data interpretation are superficial. Furthermore, several key aspects—including appropriate controls, proper contextualization within the existing literature, and mechanistic insights—are either lacking or insufficiently developed.

At this stage, I do not recommend the manuscript for publication. A substantially revised version would be necessary, one that addresses the following major concerns in both experimental design and data interpretation.

Major Concerns

Genetic and Radiological Context Lacking
The manuscript does not specify whether any of the cases—especially those with multiple cavernomas (14.2% of the cohort)—represent familial forms associated with known mutations in CCM1, CCM2, or CCM3. This is a significant omission, as loss-of-function mutations in these genes are known to drive inflammatory signaling, including COX-2 upregulation (PMID: 24291398 ; 27639680).

Response: Dear valued Reviewer. Thank you for pointing out this well-known correlation between CCM1-3 mutation and loss-of-function mutations, which drive inflammatory signaling. In our study cohort, only one patient presented a CCM1 mutation. The connection between genetic mutations and increased inflammatory cascades remains a very interesting topic, which sadly we cannot answer with this study population due to the available n=1. To emphasize that this topic is nevertheless relevant and should continue to be considered in future studies, we have referred to the CCM1 population in our cohort in the section (Results/Study cohort). Furthermore, we have added a section to the discussion that highlights the connection and continues to refer to the need to consider this in future work (Discussion).

Furthermore, no reference is made to developmental venous anomalies (DVAs), which are frequently observed in sporadic CCM cases and have been implicated in lesion formation via inflammatory pathways (PMID: 19833796 ; 36498972). Including these clinical and radiological aspects would provide important context for interpreting the inflammatory findings.

Response: Thank you very much for pointing out the pathological connection between DVAs and CCM in general. Based on your suggestion, we have adapted the manuscript and statistically analyzed patients with DVAs. Changes to the manuscript have been made in the following aspects:

Methods/Radiological data collection and measurements

Results/Study cohort – end of first paragraph.

Results/CCM and its association with NLRP3

Results/CCM and its association with COX-2

Antibody Validation and Reproducibility

The differences observed in COX-2 and NLRP3 protein expression may be influenced by varying sensitivity and specificity of the primary antibodies used. The authors do not address this possibility. Validation using additional antibodies or complementary techniques (e.g., Western blotting or qPCR) is recommended to confirm the reproducibility and reliability of the IHC results.

Response: Thank you very much for pointing out ways to improve and strengthen the methodology used in this paper. The antibodies used in this study (COX-2: 1:50 dilution, DAKO, Clone: CX-294; NLRP3: 1:200 dilution, ABCAM, ab214185) have already been used in many studies and are well established in terms of their specificity and sensitivity. Further verification of these results by the manufacturers and other scientists is always desirable in order to strengthen our own methodology. Due to limited financial resources, we were unable to obtain further confirmation in this study. Nevertheless, we added this to our limitations. Changes in the manuscript were made in “Limitations”.

Lack of Cell-Type-Specific Analysis
The IHC quantification appears to be performed over entire tissue sections, without differentiating between distinct cell populations (e.g., endothelial cells, glial cells, inflammatory cells). This limits the biological relevance of the data. A cell-specific analysis, possibly using co-staining or segmented image analysis, would enhance the interpretability of protein expression patterns and their functional implications.

Response: Thank you for your suggestion to improve the readability and interpretability of the results shown here. Based on this, we have discussed the morphology and analyzed cells in detail in the first section of the discussion to ensure better understanding and reproducibility. The cell-specific evaluation of enzyme expression that you propose is an intriguing method that will certainly add value to future work. However, this work is not only about the cavernoma tissue, but also about the microenvironment surrounding the CCM tissue. We therefore decided to analyze the complete image sections. We have now specified this again in the first section of the discussion. Changes have been made to first section in discussion.

Inappropriate Interpretation of Semi-Quantitative IHC
The IHC analysis conducted only allows for semi-quantitative assessment of protein expression. Therefore, references to enzymatic expression rates (g., Figure 3) or inflammatory pathway activation are not only inappropriate but also potentially misleading.
In particular, NLRP3 is not an enzyme but a component of the inflammasome complex, whose activation involves intricate regulatory mechanisms, including NLRP3 post-translational modifications and interactions with other proteins, such as ASC and caspase-1, which ultimately leads to cytokine activation (e.g., IL-1β, IL-18; PMID: 36741414 ). Similarly, COX-2 has complex, context-dependent roles in both physiological and pathological processes (PMID: 9737710 ), and IHC alone does not provide insights into its functional activity or downstream effects.
Thus, without measuring the levels of downstream effectors, such as COX-2-derived prostanoids (e.g., prostaglandin E2) or NLRP3-inflammasome-related cytokines (e.g., IL-1β, IL-18), the conclusions drawn about inflammatory pathway activation are speculative and unsupported.

Response: Thank you again very much for pointing out ways to improve and strengthen the methodology used in this paper. As mentioned above due to financial limitations and lack of further CCM tissue samples of all patients, we were and are not able to conduct any further experiments. As we already mentioned a multicentric registry with tissue samples would improve future research in this area substantially. For this work we sadly are not able to provide any further experiments.

Changes have been made to:

Limiations

Conclusion

Titel Figure 3.

Oversimplified and Unsupported Conclusions
The conclusion that “the upregulated inflammatory pathways suggest a dominant role of both shear stress (COX-2) and ischemia-induced inflammation (NLRP3)” (Section 6, lines 313–315) is speculative and reductive. Both COX-2 and NLRP3 are responsive to a wide array of stimuli, including oxidative stress, metabolic shifts, and pathogen- or damage-associated molecular patterns. Assigning their activation solely to shear stress or ischemia, in the absence of mechanistic evidence, is an overinterpretation.

Response: Thank you very much for discussing this very central point of this work again. We have adjusted the discussion and conclusion to reflect the fact that the individual enzymes are not only responsible for mechanical or ischemic triggers.

Overlooked Crosstalk Between COX-2 and NLRP3
The manuscript treats COX-2 and NLRP3 as independent inflammatory markers, yet increasing evidence indicates a reciprocal relationship between them. COX-2 can influence NLRP3 inflammasome activation, and NLRP3 signaling can, in turn, regulate COX-2 expression (e.g., PMID: 25294243 ; 39832004; 28628921). This crosstalk is mechanistically relevant and should be acknowledged in the discussion, as it may affect the interpretation of their co-expression in CCM tissues.

Response: Thank you very much for discussing this very central point of this work again. We already discussed it in a part of the discussion (lines 280-292 of the first submitted manuscript, including 25294243 ). We have now added upon your proposal the work of Zhuang et al to the discussion, which supports our findings. Changes have been made to the Discussion.

Failure to Cite Prior Foundational Studies
The upregulation of COX-2 in CCM lesions has been originally demonstrated and discussed in previous reports (PMID: 24291398 ; 27639680), yet these studies are not cited. Proper citation of foundational literature is essential to acknowledge prior work and to contextualize the current findings within the evolving understanding of inflammation in CCM pathology.

Response: Thank you for pointing out the lack of context in the paper. We have once again searched the literature for the fundamental pioneers and specifically incorporated the works you mentioned into the paper (Introduction/Discussion). Recognition of prior and pioneering work remains an integral part of research.

Additional Comments

Consider including additional controls or validation experiments to strengthen the reliability of the immunohistochemical findings.

Response: as mentioned above.

Ensure that all references to molecular pathways and protein functions are consistent with current literature and mechanistic understanding.

Response: We have adapted the introduction, discussion, and references to the current state of research.

Including a schematic diagram summarizing the hypothesized inflammatory mechanisms (based on both current and prior findings) might help readers visualize the proposed model more effectively.

Response: Thank you for suggesting a schematic overview of the hypothesized inflammatory mechanisms. The work itself does not imply any pathophysiological pathways of the inflammatory cascade that are not already known. We have therefore decided to refer to pioneering work and previous studies on inflammatory mechanisms rather than generating our own representational diagram.

Thank you very much for your time and effort in reviewing this work.

Reviewer 3 Report

Comments and Suggestions for Authors

The manuscript by Rodemerk et al “Inflammation in cerebral cavernous malformations: Differences between malformation related epilepsy vs. symptomatic hemorrhage” reports the differential expression of two markers of inflammation in CCM samples from patients with either epilepsy or symptomatic hemorrhage. They report expression of COX2 and NLRP3, major inflammatory regulators, in both conditions, with hemorrhage samples consistently higher than epilepsy samples. Overall, the impact of the study is low, as inflammation has already been linked to CCM. The difference between epileptic samples and hemorrhage samples is interesting, but lacks adequate controls, furthermore no mechanistic explanation is offered for this difference.

Major Concerns:

  1. Patients were grouped based on clinical MRI data into hemorrhage and epilepsy (without hemorrhage). It is not indicated how long after MRI the lesions were surgically removed- if a longer period it is possible that bleeding may have occurred. The authors should perform staining on the CCM samples for hemosiderin to prove that the epilepsy samples had no bleeding.
  2. How the five regions of each CCM analyzed were chosen should be detailed in the methods.
  3. There is a very high background in the DAB staining of the CCM samples, but not the positive controls. This could be partially ameliorated by appropriate white balance when images are captured. Negative controls should use serial sections and should also be included in Figure 2.
  4. As positive controls are not brain tissue, normal brain should also be included as a control.
  5. The phrase (line 227) ”…epilepsy was negatively linked to higher enzymatic expression of COX-2…” is inaccurate as a negative correlation is not established.

Minor Concerns

  1. Figure 2 needs more extensive labeling to indicate what was stained for, and needs negative controls.
  2. Additional inflammatory markers have been identified in CCM. Whether these are also differentially expressed would add to the manuscript.
  3. The authors did not cite several publications that have linked inflammation to CCM. Specifically, Goitre et al Free Radical Biology and Medicine 2014, which first identified COX2, should be included.
  4. The authors oversimplify the relationship between COX2 and NLRP3 in the discussion and abstract as being “mechanical” and “ischemic” when in fact both molecules can be active in both contexts.
Comments on the Quality of English Language
  1. There are several phrases that are consistently misused. The phrase “against patients” is used in several places. [line 311: This study identified upregulated enzyme expression of both COX-2 and NLRP3 in patients who had as the first symptom a cerebral hemorrhage against patients with CCM-related epilepsy.] This should be revised to use “compared to” or “versus” in place of “against”.
  2. Graph labels in Figure 3 are misspelled.

Author Response

The manuscript by Rodemerk et al “Inflammation in cerebral cavernous malformations: Differences between malformation related epilepsy vs. symptomatic hemorrhage” reports the differential expression of two markers of inflammation in CCM samples from patients with either epilepsy or symptomatic hemorrhage. They report expression of COX2 and NLRP3, major inflammatory regulators, in both conditions, with hemorrhage samples consistently higher than epilepsy samples. Overall, the impact of the study is low, as inflammation has already been linked to CCM. The difference between epileptic samples and hemorrhage samples is interesting, but lacks adequate controls, furthermore no mechanistic explanation is offered for this difference.

Major Concerns:

    Patients were grouped based on clinical MRI data into hemorrhage and epilepsy (without hemorrhage). It is not indicated how long after MRI the lesions were surgically removed- if a longer period it is possible that bleeding may have occurred. The authors should perform staining on the CCM samples for hemosiderin to prove that the epilepsy samples had no bleeding.

Response: Thank you for pointing out the methodological weakness. Prussian blue staining was already performed on all patients as part of routine clinical practice, but compared to the MRI examination before surgery, there was no difference in the final classification of the patients. In order to reinforce the distinction made in this study between patients with symptomatic bleeding and cavernoma-dependent epilepsy, we have now included the results of the pathological examination in the first paragraph of the results section. “Results/Study cohort.”

    How the five regions of each CCM analyzed were chosen should be detailed in the methods.

Response: To ensure the reproducibility of the study, we have specified the criteria for image selection in the Methods/Histological Image Analysis section.

    There is a very high background in the DAB staining of the CCM samples, but not the positive controls. This could be partially ameliorated by appropriate white balance when images are captured. Negative controls should use serial sections and should also be included in Figure 2.

Response: We have improved the white balance in the CCM Tissue sample pictures in Figure 1. The first impression of high DAB background staining was linked to poor colour grading in the initial image acquisition. This has been resolved.

Negative controls are not methodologically necessary for every sample and therefore redundant for Figure 2. As part of the functional testing and repeatability (shown in Figure 1 of the paper), it was clearly demonstrated that the antibody binds specifically and that DAB shows negligible background staining in the staining method.

For this reason, and also to prevent cluttered graphics, we will not include any further negative controls in the figures.

    As positive controls are not brain tissue, normal brain should also be included as a control.

Response: A positive control with healthy brain tissue is not methodologically necessary. A positive control in the context of antibody establishment is intended to demonstrate that the antigen can be stained in a tissue using an applied staining method. This has been sufficiently demonstrated in this study. The added value of any other control of “normal” brain tissue for COX-2 and NLRP3 does not add value to the objective of this study. Furthermore, the examination of healthy human brain tissue is ethically very questionable.

    The phrase (line 227) ”…epilepsy was negatively linked to higher enzymatic expression of COX-2…” is inaccurate as a negative correlation is not established.

Response: Thank you for pointing out the misinterpretation. We have carefully adjusted the interpretation those results (End of section Results/CCM and its association with COX-2).

Minor Concerns

    Figure 2 needs more extensive labeling to indicate what was stained for, and needs negative controls.

Response: For better understanding, we have adjusted Figures 1 and 2. Negative controls are not methodologically necessary for every sample, but only when establishing the methodology. Since no background staining by the first antibody can be seen in the negative controls already shown in Figure 1, further presentation of negative controls is redundant and unnecessary.

    Additional inflammatory markers have been identified in CCM. Whether these are also differentially expressed would add to the manuscript.

Response: The detection of further inflammatory markers is certainly conducive to understanding the pathophysiology of cavernomas. Unfortunately, it is no longer possible to include further immunohistochemical staining in this study, as many tissue samples have been completely used up. In order to address this problem of small sample sizes and limited sample material, we already refer in the conclusion of this paper to the goal of a multicenter registry, which would allow us to conduct these further investigations and go into much greater detail.

    The authors did not cite several publications that have linked inflammation to CCM. Specifically, Goitre et al Free Radical Biology and Medicine 2014, which first identified COX2, should be included.

Response: Thank you for pointing out our error in citing the pioneer work. We have included several more prior works in this field.

    The authors oversimplify the relationship between COX2 and NLRP3 in the discussion and abstract as being “mechanical” and “ischemic” when in fact both molecules can be active in both contexts.

Response: Thank you very much for discussing this very central point of this work again. We have adjusted the discussion and conclusion to reflect the fact that the individual enzymes are not only responsible for mechanical or ischemic triggers.

Comments on the Quality of English Language

    There are several phrases that are consistently misused. The phrase “against patients” is used in several places. [line 311: This study identified upregulated enzyme expression of both COX-2 and NLRP3 in patients who had as the first symptom a cerebral hemorrhage against patients with CCM-related epilepsy.] This should be revised to use “compared to” or “versus” in place of “against”.

Response: Thank you very much for pointing out the orthographic improvements. We revised the manuscript accordingly.

    Graph labels in Figure 3 are misspelled.

Response: Thank you for bringing this orthographic error to our attention. We have now adjusted the graphs accordingly.

Thank you very much for your time and effort in reviewing this work.

Round 2

Reviewer 3 Report

Comments and Suggestions for Authors

The authors have made significant improvements to the manuscript. While I do not agree that normal controls would not be informative, the conclusions are sufficiently precise to allow the paper to proceed. 

I did detect a few typos while reviewing this version that should be corrected prior to publication.

Author Response

The authors have made significant improvements to the manuscript. While I do not agree that normal controls would not be informative, the conclusions are sufficiently precise to allow the paper to proceed. 

I did detect a few typos while reviewing this version that should be corrected prior to publication.

Answer: Thank you very much for your time and effort in reviewing this article. We have checked the manuscript again for spelling mistakes.